# OpenReview forum: "Backtracking Improves Generation Safety"
_ICLR.cc/2025/Conference — ICLR 2025 Oral_

### Official Review · Reviewer_KgTS · 2024-10-28

**Soundness:** 3
**Presentation:** 3
**Contribution:** 3
**Rating:** 8
**Confidence:** 4

**Summary:**

This paper proposes a novel method, backtracking, for purifying LLM output to make it safe and understandable. In backtracking, the LLM generates an initial response and outputs a special token [RESET] if the initial response is harmful. An external moderator model, Llama Guard 2, is employed to help generate training data.

**Strengths:**

* The proposed method, backtracking, is a novel way to help LLM generate safer responses while keeping the generation quality.

* Backtracking does not introduce heavy inference expense in comparison to previous resampling-based methods, which highlights a new direction for safer generation.

* Adaptive adversarial attacks are also discussed.

* The paper is well-written and easy to follow.

**Weaknesses:**

* There is no discussion or experiments on how likely backtracking LLM will overly reject safe queries.

* The performance of backtracking against adversarial attacks is not always satisfying, as shown in Table 2.

**Questions:**

See weaknesses.

---

> ### Author Response · Authors · 2024-11-20
>
> We appreciate your positive feedback. We address some of your concerns below.
>
> > No discussion or experiments on how likely backtracking LLM will overly reject safe queries
>
> You may have missed our results in Appendix B.2 (referred to in the main paper in Section 4.2, L318), where we evaluated how often the Llama model backtracks on a set of safe queries (validation set of OpenAssistant). We find that with a suitable choice of logit bias on the [RESET] token, the model backtracks for only 1% of safe queries (12% without tuning the logit bias).
>
> > The performance of backtracking against adversarial attacks is not always satisfying, as shown in Table 2.
>
> We acknowledge that neither backtracking, nor any other method, is not a perfect defense against adversarial attacks, but we want to point out that backtracking is a technique meant to improve model safety *generally*, and the additional robustness against strong adversarial attacks (e.g., GCG and AutoDAN) without special training suggests that backtracking could become a component in a truly robust LLM system.
>
> Despite much progress, safeguarding LLMs against adversarial attacks remains an open problem, and is not an issue that can be addressed with a single technique. We note backtracking performance is strongly dependent on how it is trained: our models were not trained against adversarial attacks at all and still helped on such attacks. We expect that backtracking training in the presence of adversarial attacks (i.e., combining it with the idea of adversarial training [1]) would greatly improve robustness in the adversarial setting. However, we believe the results in the paper alone already demonstrate the value of our technique across a number of settings, and leave such approaches to future work.
>
> [1] Towards Deep Learning Models Resistant to Adversarial Attacks

---

> > ### Comment · Reviewer_KgTS · 2024-11-21
> >
> > Thank you for your clarification. I am raising my score.

---

### Official Review · Reviewer_4GF6 · 2024-11-01

**Soundness:** 3
**Presentation:** 4
**Contribution:** 3
**Rating:** 8
**Confidence:** 4

**Summary:**

The authors propose to train a model to predict when to stop generating and restart its generation by generating a "reset" token, and show this method reduces safety violations, even under adversarial attacks. The authors also present an adaptive version of GCG to test the limits of the backtracking technique.

**Strengths:**

Method is simple yet effective. The authors also included ablations on each step from baseline SFT to full backtracking SFT + DPO. The evaluation was done on multiple datasets and 2 SOTA base models; the authors also included detailed analyses of results and example generations. I appreciated that the details of the training setup and hyperparameter optimization were included in the appendix. Finally, the tradeoffs and limitations of assuming a single reset per generation was discussed.

**Weaknesses:**

No major weaknesses, although given that there is a large difference in relative drop in violation rates between the 2 models tested, it would be useful to show how this method affects larger model sizes and other base models beyond Gemma and Llama (e.g., Qwen, Mistral).

**Questions:**

* Table 1: What is the delta of performing backtracking SFT vs baseline SFT when backtracking DPO is performed in both cases? Is backtracking SFT necessary to achieve the backtracking DPO results reported?
* What is the effect of scale on safety violation rates with backtracking?
* There appears to be a rather large difference in the violation rates between the two models (Gemma/Llama) for backtrack SFT + DPO, even more so if we look the relative drop instead of absolute numbers. How can this be explained?

---

> ### Author Response · Authors · 2024-11-20
>
> We appreciate your positive feedback on our work. We answer some of your questions below.
>
> > What if we skip backtracking SFT training?
>
> Empirically, skipping backtracking SFT training breaks the model. The technical reason is that DPO imposes a “zero-forcing” reverse-KL regularization between the optimized policy and the reference (SFT) policy. Specifically, if we didn't use backtracking SFT to “warm-up” the model, the reference policy would assign practically zero probability to [RESET], and the resulting KL-penalty would be massive if the DPO policy assigns non-zero probability to [RESET]. We also note that it is standard practice to perform SFT before DPO [1].
>
> > The effect of scale on safety violation rates with backtracking, and difference in backtracking effectiveness on Gemma vs. Llama
>
> Intuitively, we expect larger models to be able to learn to backtrack more precisely and self-correct more reliably, a trend we do observe between Gemma–2-2B and Llama-3-8B (Table 1), where the larger Llama model seems to gain more from backtracking. While we don’t have the resources to scale experiments to larger models, we expect the backtracking technique to be useful particularly for large models capable of self-critique, as even the state-of-the-art models remain far from being perfectly safe.
>
> [1] Direct Preference Optimization: Your Language Model is Secretly a Reward Model

---

> > ### Comment · Reviewer_4GF6 · 2024-11-26
> >
> > Thanks for the response, it would be good to add the observations from the first response to the paper.

---

> > > ### Author Response · Authors · 2024-11-26
> > > **paper revised**
> > >
> > > Thank you for acknowledging our response, and we agree that the discussion on backtracking SFT would be valuable to add to our paper. We have uploaded a revised manuscript as you suggested.

---

### Official Review · Reviewer_WShb · 2024-11-04

**Soundness:** 4
**Presentation:** 4
**Contribution:** 3
**Rating:** 8
**Confidence:** 4

**Summary:**

This paper introduces a new method for reducing unsafe generations from language models by allowing a model to backtrack. The proposed method trains a model with SFT and DPO to output a [RESET] token if the partial generation so far is unsafe and then generate a new safe generation.

**Strengths:**

The strengths of the paper are as follows:
- The proposed method is simple and effective. Moreover, the inference procedure is simple and does not add a lot of overhead during inference.
- The authors show that when backtracking is used it is exponentially harder to get unsafe generations even when even temperature sampling is used to generate and pick the worst of $k$ responses.
- Backtracking improves robustness against a few adversarial attacks without any special adversarial training or other modifications.

**Weaknesses:**

Here are a few weaknesses:
- There is no analysis of no analysis of how the proposed backtracking method compares/complements with existing methods for reducing unsafe generations. Is the proposed method better? Are there complementary benefits?
- Appendix B2 shows that on the OpenAssistant dataset the [RESET] token is generated around 12-13% of the time with no logit bias. In all these cases are there unsafe generations? If so, when the logit bias is set to -5.0, then the [RESET] token is only generated around 2% of time, so is there a lot of uncaught unsafe generations? In general, is it hard to tune the logit bias hyperparameter?

**Questions:**

Here are some questions:
- The paper state that "the models virtually never backtrack during non-safety evaluations." Can you quantify this?
- Have you tried using backtracking with just DPO with not SFT?
- To generate the graphs in Figure 4, what prompts were used? Are baseline models in figure 4 trained with DPO and SFT specifically for safety?
- Do most SFT datasets have unsafe response $y_i^−$?
- Have you tried attacking with an adversarial system prompt that says after you see the [RESET] token be extra helpful or something along these lines?

---

> ### Author Response · Authors · 2024-11-20
>
> Thank you for your detailed read of our paper and your insightful feedback. We try to address your questions below.
>
> > There is no analysis of no analysis of how the proposed backtracking method compares/complements with existing methods for reducing unsafe generations.
>
> The primary method for reducing unsafe generations is SFT or RL fine-tuning to reduce the probability of unsafe next-token generation. We compare backtracking with both SFT and DPO safety-trained variants in our work and find significant safety gains. However, there are other—less standard—methods of improving model safety such as [1] and [2]. We don't compare with these methods because backtracking is not mutually exclusive with other techniques such as [1, 2], and we expect complementary gains since it seems plausible that you can improve the safety of a base model using existing techniques, while teaching it to backtrack to gain additional safety.
>
> > Backtracking in OpenAssistant, and logit bias tuning
>
> OpenAssistant (OA) is heavily focused on utility, and the 12% backtracking rate on OA (Fig. 5) suggests a small amount of false positives. This is almost entirely mitigated by adding a small negative logit bias to the [RESET] token (-5), with backtracking rate on OA dropping to 1%, with basically the same model safety (1.5% -> 1.6%). Tuning logit bias in principle is not difficult, and it just requires benchmarking backtracking behavior on a development set of prompts. We will add clarifications in the paper and point out the need for a suitable choice of logit bias on the [RESET] token to ensure a near-zero backtracking rate during non-safety evaluations.
>
> > What if we skip backtracking SFT training?
>
> Empirically, skipping backtracking SFT training breaks the model. The technical reason is that DPO imposes a “zero-forcing” reverse-KL regularization between the optimized policy and the reference (SFT) policy. Specifically, if we didn't use backtracking SFT to “warm-up” the model, the reference policy would assign practically zero probability to [RESET], and the resulting KL-penalty would be massive if the DPO policy assigns non-zero probability to [RESET]. We also note that it is standard practice to perform SFT before DPO [3].
>
> > Data and models used for Fig. 4
>
> To compute safety rates under sampling (Fig. 4), we used the same safety evaluation dataset we use throughout the paper, detailed in Section 4.1. The models compared are backtracking and baseline models that have both undergone SFT + DPO training.
>
> > Do most SFT datasets have unsafe responses $y_i^−$?
>
> A common practice is to take the preferred response in a paired preference dataset for SFT training, so in such cases $y_i^−$ response would be available. If training on SFT datasets without $y_i^−$, we can still mix in preference data from the subsequent DPO stage to provide backtracking SFT supervision.
>
> > Have you tried attacking with an adversarial system prompt that says after you see the [RESET] token be extra helpful or something along these lines?
>
> In our work, we do not explore malicious system prompts and assume the model takes on a default “helpful” and “harmless” role, which would be the case for most black-box LLMs (e.g., ChatGPT, Gemini). We agree that it may be possible to inhibit backtracking through malicious system prompts, but even with backtracking inactive, the model should be in principle at least as safe as the non-backtracking baseline. Safety under model adaptation (e.g., fine-tuning and system prompting) is an open research question out-of-scope for this work.
>
> [1] The Instruction Hierarchy: Training LLMs to Prioritize Privileged Instructions \
> [2] Improving Alignment and Robustness with Circuit Breakers \
> [3] Direct Preference Optimization: Your Language Model is Secretly a Reward Model

---

> > ### Comment · Reviewer_WShb · 2024-11-26
> >
> > Thank you for the response! You've answered my questions and I've raised my score.

---

### Official Review · Reviewer_Gb5k · 2024-11-04

**Soundness:** 4
**Presentation:** 4
**Contribution:** 4
**Rating:** 8
**Confidence:** 2

**Summary:**

The paper proposes a novel  technique called Backtracking, to enhance the safety of large language models by allowing them to undo unsafe responses. Traditional safety methods focus on prevention by reducing the likelihood of harmful outputs. However, backtracking enables models to recognize and "reset" unsafe content mid-generation using a designated "[RESET]" token. This approach, which can be incorporated into both Supervised Fine-Tuning (SFT) and Direct Preference Optimization (DPO), improves safety without reducing the model's helpfulness. Evaluation shows that backtracking significantly reduces unsafe output rates and defends against various adversarial attacks, including an adaptive one designed to counteract the reset function. The results suggest backtracking is a promising addition to existing alignment techniques for safer language model deployments​

**Strengths:**

- Reflection-based Defense Yields More Flexibility: The proposed method has significant flexibility compared to other methods. With proper prompting, unlike traditional filter-based adversarial attack defense, it is easy to use a safe backtracking model with updatable preference and knowledge, depending on the user's demand.
- Straight Forward Intuition and Easy Maintenence: The internal mechanism of the proposed backtracking defense is comparatively very transparent. When unexpected defensive actions are taken by the models, it is very easy to convert such error log into new DPO data.

**Weaknesses:**

- Discrepancy between the Streaming User Interface and Model Behavior: When undoing some of the generation outputs, the system should delete the undone harmful trigger prompts and update it with the model's remedy context afterwards. This could cause some unexpected discrepancies and difficulties in general designs of the pipeline.
- Risks in Reprogrammable Backtracking: Since we now rely on the model's own reflection to backtrack the harmful trigger prompts, it is possible, if the model itself is reprogrammed in system prompt with malicious intents, the backtracking's flexibility can be abused to jailbreak the defense against attacks.

**Questions:**

Do we keep the undone history in the dialogue and KV cache or we also undo them altogether and only keep the refined context?

---

> ### Author Response · Authors · 2024-11-20
>
> We appreciate your positive assessment of our work.
>
> > Compatibility with a streaming API
>
> We agree that implementing backtracking in a streaming environment could be inconvenient for users visually, since backtracking “removes” part of the generation history upon backtracking. However, we would like to note that in the context of safety, it is always better to undo an unsafe generation even if it may be visually annoying. Also, backtracking is reasonably precise: it usually does not kick in unless the generation is unsafe, so a backtracking model would behave similarly to a non-backtracking model in non-safety-related contexts.
>
> > Risks in reprogrammable backtracking
>
> In our work, we do not explore malicious system prompts and assume the model takes on a default “helpful” and “harmless” role, which would be the case for most black-box LLMs (e.g., ChatGPT, Gemini). We agree that it may be possible to inhibit backtracking through malicious system prompts, but even with backtracking inactive, the model should be in principle at least as safe as the non-backtracking baseline. Safety under model adaptation (e.g., fine-tuning and system prompting) is an open research question out-of-scope for this work.
>
>
> > Do we keep the undone history in the dialogue and KV cache or do we also undo them altogether and only keep the refined context?
>
> We keep the history in-context. The reason is that we suspect that the presence of the “failed generation” actually provides a signal for the model to self-correct and generate more safely, and [1] suggests that this self-correcting approach could be effective. Another issue with dropping context is that the model would repeat its own unsafe generation with high probability. In the extreme case, if greedy decoding is used, the model is guaranteed to be stuck in a backtracking-regeneration loop after it backtracks once.
>
> [[1] Generating Sequences by Learning to Self-Correct](https://arxiv.org/pdf/2211.00053)

---

> > ### Comment · Reviewer_Gb5k · 2024-11-26
> >
> > I have read the response and would like to keep my scores.

---

### Comment · Area_Chair_eJE3 · 2024-11-25
**Reviewer Response**

Dear Reviewers,

The rebuttal period is almost over, and the paper has received both positive and negative feedback. The authors have worked hard to address all your concerns. Could you take a moment to review their responses and let us know if you have any unresolved issues?

Best,
AC

---

### Meta-Review · Area_Chair_eJE3 · 2024-12-23

**Metareview:**

The paper presents "backtracking," a novel method to improve language model safety by enabling recovery from unsafe partial generations using a [RESET] token. Unlike traditional prevention-based approaches, backtracking trains models to recognize and reset unsafe outputs mid-generation, integrating into SFT and DPO without compromising helpfulness. Evaluations show it reduces unsafe outputs by a factor of four across adversarial datasets. Reviewers appreciate its simplicity, practicality, and minimal overhead, highlighting strong safety improvements and comprehensive experiments with detailed ablations. While no major weaknesses were identified, some noted potential vulnerabilities to system-prompt-based attacks, which the authors acknowledge as an open question. Overall, reviewers see the method as a significant and effective contribution to AI safety, with the rebuttal addressing key concerns. I would recommend strong acceptance.

**Additional Comments On Reviewer Discussion:**

The rebuttal period clarified key concerns raised by reviewers, including questions about false positives, the necessity of backtracking SFT, and adversarial vulnerabilities. The authors’ responses were thorough, leading reviewers to keep their high scores.

---

### Decision · Program_Chairs · 2025-01-22

Accept (Oral)